# Exosomes and Hepatocellular Carcinoma: From Bench to Bedside

**DOI:** 10.3390/ijms20061406

**Published:** 2019-03-20

**Authors:** Reina Sasaki, Tatsuo Kanda, Osamu Yokosuka, Naoya Kato, Shunichi Matsuoka, Mitsuhiko Moriyama

**Affiliations:** 1Department of Gastroenterology and Nephrology, Chiba University, Graduate School of Medicine, 1-8-1 Inohana, Chuo-ku, Chiba 260-8670, Japan; reina_sasaki_0925@yahoo.co.jp (R.S.); yokosukao@faculty.chiba-u.jp (O.Y.); kato.naoya@chiba-u.jp (N.K.); 2Division of Gastroenterology and Hepatology, Department of Medicine, Nihon University School of Medicine, 30-1 Oyaguchi-kamicho, Itabashi-ku, Tokyo 173-8610, Japan; matsuoka.shunichi@nihon-u.ac.jp (S.M.); moriyama.mitsuhiko@nihon-u.ac.jp (M.M.)

**Keywords:** exosome, hepatocellular carcinoma, microRNA, long noncoding RNA

## Abstract

As hepatocellular carcinoma (HCC) usually occurs in the background of cirrhosis, which is an end-stage form of liver diseases, treatment options for advanced HCC are limited, due to poor liver function. The exosome is a nanometer-sized membrane vesicle structure that originates from the endosome. Exosome-mediated transfer of proteins, DNAs and various forms of RNA, such as microRNA (miRNA), long noncoding RNA (lncRNA) and messenger RNA (mRNA), contributes to the development of HCC. Exosomes mediate communication between both HCC and non-HCC cells involved in tumor-associated cells, and several molecules are implicated in exosome biogenesis. Exosomes may be potential diagnostic biomarkers for early-stage HCC. Exosomal proteins, miRNAs and lncRNAs could provide new biomarker information for HCC. Exosomes are also potential targets for the treatment of HCC. Notably, further efforts are required in this field. We reviewed recent literature and demonstrated how useful exosomes are for diagnosing patients with HCC, treating patients with HCC and predicting the prognosis of HCC patients.

## 1. Introduction

There are at least three extracellular vesicles in the extracellular microenvironment, including exosomes, microvesicles and apoptotic bodies [1]. The exosome is a nanometer-sized (30–100 nm) membrane vesicle structure that originates from the endosome. Exosomes are secreted from almost all cells from diverse organisms [2]. The composition of exosomes resembles parental cells, and exosomes provide a circulating traceable specific signature. Exosomes are thought to be key players in numerous biological processes in both normal and pathologic contexts [2]. Evidence indicates that proteins, DNAs and various forms of RNA, such as microRNA (miRNA), long noncoding RNA (lncRNA) and messenger RNA (mRNA) transferred by exosomes, contribute to the development of cancers, neurodegenerative disorders, collagen diseases and other pathological conditions (Figure 1) [3,4,5,6]. Nonetheless, the physiologic functions of exosomes remain largely unknown.

Cancer-derived exosomes form network complexes of communication between tumor and nontumor cells. On the one hand, exosome-mediated cancer progression through the promotion of a tumor microenvironment, such as enhancing cell proliferation [7,8] and an epithelial-mesenchymal transition (EMT) [8], increases of tube formation in angiogenesis [9] and the ability to escape from immune surveillance [10]. On the other hand, cancer-derived exosomes play a critical role in immunomodulators [11]. These immunomodulators may cause a strong immune response.

Hepatocellular carcinoma (HCC) is the fourth most common cancer type and the second most common cause of cancer-related deaths [12]. HCC is mainly associated with chronic liver disease and cirrhosis [13]. Recently, therapeutic options for hepatitis C virus (HCV) and hepatitis B virus (HBV), which are responsible for chronic liver diseases, have been developed such as direct-acting antiviral agents (DAAs) and nucleos(t)ide analogues, respectively [14,15]. Universal HBV vaccination has also been introduced in many countries [15]. However, obesity leading to non-alcoholic liver disease (NAFLD) and non-alcoholic steatohepatitis (NASH) is increasing as one of the major etiologies for liver diseases and HCC [16]. The treatment options for advanced-stage HCC are very limited. HCC can be treated with liver transplantation, surgical resection, radiofrequency ablation (RFA), transcatheter arterial chemoembolization (TACE) and systemic chemotherapy [13]. Among these options, liver transplantation and surgical resection seem to be potential curative approaches. However, some patients drop out from liver transplantation because of the increasing waiting time and the shortage of donor organs [13].

This review focuses on the contents of exosomes and how exosomes contribute to the development of HCC. We further address how useful exosomes are to diagnose HCC, to treat patients with HCC and to predict their prognosis.

## 2. HCC

HCC largely occurs in the background of chronic liver disease and cirrhosis in over 90% of cases [13]. In the US and Japan, HCV is the major cause of chronic hepatitis, cirrhosis, and HCC, while in Asian and African countries, HBV is a common cause for these diseases [15]. Other etiologies, such as alcoholic liver disease, NAFLD and NASH, also contribute to chronic liver diseases and HCC. Recently, DAAs have been shown to eradicate HCV from over 90% of chronically HCV-infected patients [14]. As of 2012, 183 countries initiated an HBV vaccination program for newborn infants, which decreased HBV-associated liver diseases [15]. However, obesity leading to NAFLD and NASH is increasing as one of the leading etiologies for liver diseases and HCC [17].

The high mortality rates of HCC are almost equal to the incidence rates of HCC in most countries, indicating the lack of effective therapies [13]. The treatment options for advanced HCC are very limited, although there are several treatment options for HCC as follows: liver transplantation, surgical resection, RFA, TACE and systemic chemotherapy [13,18]. Moreover, the recurrence rate of HCC after surgical resection is 30–70% within 5 years [19,20,21].

During the last decade, the oral multitargeted tyrosine kinase inhibitor sorafenib was first approved as an oral agent of systemic chemotherapy in patients with advanced and metastatic HCC [22]. However, the average overall survival of patients treated with sorafenib was challenging in almost all patients with HCC [23]. Patients treated with sorafenib also acquired resistance [24] and suffered from adverse events [25]. Many trials evaluating systemic chemotherapy in patients with advanced HCC are ongoing [26,27,28].

## 3. Composition and Function of Exosomes

Exosomes are 30–100 nm vesicles with a phospholipid bilayer membrane [1,29]. The exosome is one of the membrane vesicles secreted from intracellular multivesicular bodies (MVBs) into the extracellular space by almost all cells from diverse organisms. Exosomes play a role in cell-to-cell communication in the multicellular organisms, even outside the tumor microenvironment. Moreover, cancer cells increasingly produce exosomes and transfer from cancer cells to naïve cells [30,31].

Exosomes contain a variety of cellular components, including a range of proteins such as heat shock proteins (HSPs), lipids, RNAs, mRNAs and DNA molecular cargoes, with surface protein markers, including tetraspanins (Figure 1) [32]. Exosomes are endocytic in origin and commonly contain endosome-associated proteins, including Rab GTPase, SNARE, annexin, and tetraspanins, such as CD63, CD81, CD82, CD53, and CD37. Tumor-derived exosomes are protected against degradation from enzymes since they are enclosed in a lipid bilayer [29,33]. Exosomes include some miRNA with their carriers, RNA binding proteins (RBPs), such as high-density lipoproteins and argonaute-2 (Ago2) [34]. RBPs make complexes with RNA, transport of RNAs into exosomes and maintain RNAs. However, there are no reports related to HCC-derived exosomes.

The contents of exosomes and their effects on recipient cells mainly depend on the cell types from which they are derived. It has been reported that ExoCarta (http://www.exocarta.org) is a manually curated Web-based database of exosomal proteins, RNAs and lipids [33]. Here, we discuss how HCC cells communicate with other cells in their local and distant environments using exosomes. We also focused on the contents and functions of exosomes.

## 4. Proteins in Exosomes

### 4.1. Exosome Includes Many Proteins

There are several reports that exosomes include many proteins. Using mass spectrometry analysis, He et al. reported that 213 proteins were detected in exosomes derived from HCC cell lines [35]. Interestingly, MET proto-oncogenes, receptor tyrosine kinase (MET), S100 calcium binding protein A4 (S100A4), S100A10, S100A11, caveolin 1 (CAV1) and CAV2 at high levels were included in highly malignant HCC cell-derived exosomes. They showed that MET signaling had roles in controlling cell migration and invasion, while CAV1 and CAV2 increased cell migration [35].

Moreover, S100A4 played a role in the regulation of cell motility, invasiveness and induction of metastasis by modulating cytoskeleton proteins [36]. Another group confirmed that exosomal S100A11 increased tube formation in angiogenesis and that its levels were associated with poor survival of HCC patients [9]. Moreover, both exosome and exosomal S100A11 secretion were elevated by eukaryotic translation initiation factor 3 subunit C (EIF3C) stimulation, which was up-regulated during HCC tumor development. EIF3C is up-regulated in several cancers, and it is known that silencing EIF3C induces cell apoptosis and suppresses cell proliferation and tumor growth [37]. Thus, EIF3C may be a potential target for treatment with exosome inhibitors, and reduction of S100A11 may suppress HCC angiogenesis and tumorigenesis.

Wang et al. investigated 1428 proteins in exosomes derived from HCC using mass spectrometry, and these proteins were classified by GO annotation according to their biological process, cellular component and molecular function [38]. They selected 9 oncogenic proteins, including CD44, cell division cycle 42 (CDC42), RAS related (RRAS), MET, G protein subunit alpha 13 (GNA13), a disintegrin and metalloproteinase domain 1 (ADAM1), GNAS complex locus (GNAS), eukaryotic translation initiation factor 4A3 (EIF4A3) and S100 family proteins according to ExoCarta (http://www.exocarta.org/) [33]. However, they could not identify which proteins were involved in the modification of cells into tumor-promoting cells. Thus, further studies are required at this point.

Another group identified 129 proteins that exist in exosomes derived from HCC, using protein profiling [39]. Among these proteins, a high amount of adenylyl cyclase-associated protein 1 (CAP1) was included in exosomes derived from HCC cells with high potential metastasis. CAP1 was also overexpressed in aggressive pancreatic cancer [40]. They suggested that CAP1 is associated with the metastasis and recurrence of HCC. 

Zhang et al. quantified more than 1400 exosomal proteins by performing the super-Stable Isotope Labeling using Amino Acids in Cell Culture (SILAC)-based mass spectrometry (MS) analysis on the exosomes secreted by three human HCC cell lines [41]. They reported that motile HCC cells tend to export sugar metabolism-associated proteins that are involved in glycolysis I, gluconeogenesis I and pentose phosphate pathways via exosomes.

### 4.2. HSP70

Yukawa et al. showed that exosomes derived from HCC play an important role in the influence of the immune system and angiogenesis through expressed killer cell lectin-like receptor K1 (KLRK1 /NKG2D), an activating receptor for immune cells, and HSP70, a stress-induced heat shock protein associated with angiogenesis [42]. They found that HCC-derived exosomes played an important role in the lumen formation of human umbilical vein endothelial cells (HUVECs), which was determined by the imaging of angiogenesis. It was reported that HSP70 plays roles in endothelial cell migration and lumen formation via the phosphatidylinositol 3-kinase/Akt pathway [43]. They suggested that this pathway may be a new target of treatment for HCC.

However, other groups investigated HCC-derived immunomodulators from different perspectives. Rao et al. investigated the expression of immunomodulators HCC antigens, such as HSP70, α-fetoprotein (AFP) and glypican 3, in HCC-derived exosomes [11]. Another study confirmed that with anticancer drugs, HCC-derived exosomal HSP60, HSP70, and HSP90 were up-regulated, and these HSPs, including exosomes, efficiently enhanced NK cell cytotoxicity and granzyme B production by up-regulating inhibitory receptor CD94 and down-regulating activating receptors CD69, NKG2D, and NKp44 [44]. Moreover, the resistance of anticancer drugs, such as carboplatin and irinotecan hydrochloride treatment, was associated with HSP-bearing exosome surface density and augmented cytolytic activity through NK cell-mediated cytotoxicity. These findings may contribute to the development of an efficient vaccine for HCC immunotherapy.

### 4.3. Proteins in Exosomes Play Important Roles in the Development of HCC

Fu et al. showed that attached HCC cell-derived exosomes contain SMAD Family Member 3 (SMAD3) protein, which facilitates detached HCC cell adhesion [45]. They suggested that highly malignant HCC tends to release SMAD3 through exosomes. Moreover, the abundance of SMAD3-containing exosomes positively correlated with both disease stage and pathological grade and negatively correlated with the disease-free survival of patients with HCC after surgery.

High mobility group box 1 (HMGB1) was expressed on HCC-derived exosomal membranes and bound with high affinity to Toll like receptor-2 (TLR-2), TLR-4, TLR-9 and advanced glycation end products (RAGE), which led to tumor cell survival, expansion and metastasis [10]. HMGB1 expressed on HCC-derived exosomes facilitated the production of T cell Ig and mucin domain (TIM)-1+ B cells and suppressed CD8+ T cell activity, such as proliferation and function, causing immune escape in HCC. This exosomal HMGB1-TLR-2/4-MAPK pathway may contribute to the prevention and the treatment of the immune tolerance of HCC.

Wang et al. found that the protein and mRNA levels of 14-3-3ζ are up-regulated in HCC-derived exosomes and that 14-3-3ζ impairs the anti-tumor activity of tumor-infiltrating T lymphocytes by T cell exhaustion [46]. Clinically, high expression of 14-3-3ζ in patients with HCC correlated with poor survival [47]. Wang et al. also reported that the highly expressed 14-3-3ζ group, which was distributed in highly expressed tissues by immunohistochemistry, was associated with a large tumor size, poor tumor differentiation and terminal TNM stage [46]. This study suggested a new angle for understanding T cell dysfunction in HCC [46].

Li et al. found that CXC chemokine receptor-4 (CXCR4) was elevated in high lymph node metastatic HCC-derived exosomes and promoted the migration and invasion of HCC cells with low metastatic potential [48]. A previous study showed that CXCR4 mediates matrix metalloproteinase (MMP)-9 and MMP-2 secretions to facilitate lymph node metastasis of HCC [49]. Li et al. also showed that exosomal CXCR4 promoted the proliferation rate and tube formation ability of lymphatic endothelial cells, indicating that exosomal CXCR4 might be a novel therapeutic target against tumor lymphatic metastasis [48]. Notably, inhibition of the stromal cell-derived factor-1 alpha-CXCR4 axis by CXCR4 antagonists was effective for patients with adult T-cell leukemia [50].

## 5. Noncoding RNA in Exosomes

### 5.1. MicroRNA

Sohn et al. suggested that HCC-derived exosomal miR-18a, miR-221, miR-222 and miR-224 were significantly higher and miR-101, miR-106b, miR-122 and miR-195 were lower than those in the sera from patients with cirrhosis [51]. A previous report suggested that miR-18a induced the proliferation and development of HCC in women by reducing the level of estrogen receptor-α [52]. miR-221 contributed to hepatocarcinogenesis through the dysregulation of DNA damage-inducible transcript 4 (DDIT4) [53]. miR-222 was associated with HCC cell migration through activating the AKT signaling pathway [54]. Down-regulation of miR-101 inhibited apoptosis and induced tumorigenicity by targeting myeloid cell leukemia sequence 1 (Mcl-1), an antiapoptotic member of the Bcl-2 family, in HCC [7]. miR-122 was associated with suppression of overall tumor growth and local invasion. miR-122 also regulated intrahepatic metastasis via angiogenesis in HCC [55]. Moreover, miR-122 derived from adipose tissue increased the antitumor efficacy of sorafenib on HCC in vivo [56]. miR-195 suppressed tumorigenicity and regulated the G1/S transition via modulating cyclin D1, CDK6, and E2F3 in HCC cells [57].

Thus, exosomal miRNAs may be used as novel serological biomarkers. Wang et al. showed that serum exosomal levels of miR-122, miR-148a, and miR-1246 are significantly higher in HCC than those in liver cirrhosis and normal control groups [58]. However, there were no differences between the HCC and chronic hepatitis groups. There are contradicting reports about the associations between miR-148a and HCC [59,60]. miR-1246 was reported to be overexpressed in various cancers, including HCC, but there are also contradictory reports [61,62]. One report stated that miR-1246 is oncogenic and that miR-1246 promotes cancer stemness by activating Wnt/β-actin signaling; the other report stated that miR-1246 is a tumor-suppresser and that miR-1246 inhibits cell proliferation of HCC by down-regulating nuclear factor I B (NFIB) [61,62]. Thus, they combined miR-122, miR-148a, and AFP to investigate HCC as a diagnostic marker. The area under the curve (AUC) of this combination was sufficient, and this combination can also be applied for distinguishing early HCC from liver cirrhosis.

Exosomal miR-122 is highly expressed in the liver [63]; and decreases HCC manifestation [64]. Interestingly, HCC patients with liver cirrhosis treated with TACE, who had higher miR-122 after TACE/before TACE ratio had significantly longer disease-specific survival, suggesting that the exosomal miR-122 level alterations may represent a predictive biomarker in HCC patients treated with TACE [65]. While exosomal miR-21, which is overexpressed in certain cancers, is involved in cell migration and invasion by suppressing phosphatase and tensin homology deleted on chromosome 10 (PTEN) expression [66], expression levels do not significantly change.

Fornari et al. revealed that HCC-derived exosomes mediate miR-519d, miR-21, miR-221 and miR-1228, which corelate with circulating and tissue levels [67]. They suggested that miR-519d appears to perform a better diagnostic setting of HCC than AFP, but not miR-21 and miR-221, which were run in different populations in terms of viral prevalence and ethnicity and by the means of different study approaches. A previous report showed that miR-519d promotes cell proliferation and invasion and impairs apoptosis following anticancer treatments through the direct targeting of cyclin dependent kinase inhibitor 1A (CDKN1A)/p21, PTEN, AKT serine/threonine kinase 3 (AKT3) and TIMP metallopeptidase inhibitor 2 (TIMP2) [68]. miR-519d can distinguish cirrhotic patients without HCC and cirrhotic patients with early-stage HCC.

Circulating microRNAs may be used as noninvasive biomarkers. Wang et al. found that exosomal miR-21 is significantly higher in patients with HCC compared to chronic hepatitis B patients or healthy volunteers [69]. Serum miR-21 is an independent factor for recurrence and is more sensitive than AFP [70]. Thus, differentially expressed exosomal miRNAs may act as potential biomarkers for the early detection of HCC.

Zhou et al. showed that HCC-derived exosomal miR-21 is elevated and promoted cancer progression by activating cancer-associated fibroblasts (CAFs) [71]. miR-21 converted normal HSCs to CAFs by directly targeting PTEN, led to the activation of pyruvate dehydrogenase kinase 1 (PDK1)/AKT signaling in hepatic stellate cells (HSCs) and secreted angiogenic cytokines, including vascular endothelial growth factor (VEGF), MMP-2, MMP-9, fibroblast growth factor 2 (FGF2) and transforming growth factor beta (TGFβ). Clinically, a high level of exosomal miRNA-21 correlated with CAFs, higher vessel density and survival in HCC patients. These results suggest that miR-21 may be a potential target for the prevention and treatment of HCC.

Li et al. examined 11 well-known reference genes from circulating exosomes across healthy controls, hepatitis B patients and HCC patients [72]. They found that the combination of miR-221, miR-191, let-7a, miR-181a, and miR-26a can be an optimal gene reference set for normalizing the expression of liver-specific miRNAs for comprehensive investigation into the progression of chronic hepatitis B to HCC [72]. Thus, exosomal miRNAs may be useful for monitoring hepatitis progression and as biomarkers to diagnose early stage HCC.

Sugimachi et al. found that exosomal miR-718 is significantly suppressed in patients with HCC recurrence after liver transplantation [73]. Decreased expression of miR-718 contributed to the poor prognosis of HCC patients via the up-regulation of homeobox B8 (HOXB8), and there are similar results for breast cancer patients [74]. Thus, exosomal miR-718 may be a novel biomarker for predicting HCC recurrence.

Kogure et al. identified 11 miRNAs, including miR-584, miR-517c, miR-378, miR-520f, miR-142-5p, miR-451, miR-518d, miR215-, miR-376a, miR-133b and miR-367, which are highly enriched in HCC-derived exosomes [75]. Interestingly, they identified the central involvement of these microRNAs for TGFβ-activated kinase-1 (TAK1) signaling, which is an upstream member of the mitogen-activated protein kinase kinase kinase (MAP3K) family and an essential component of cellular homeostasis and tumorigenesis in the liver by network analysis using the String 8.3 program.

Wang et al. showed that stellate cell-derived exosomes can supply miR-335-5p cargo to recipient HCC cells, inhibit HCC cell proliferation and invasion in vitro and induce HCC tumor shrinkage in vivo [76]. Another group showed that miR-335 inhibits the proliferation, migration and invasion of HCC cells via regulating the Rho-associated coiled-coil-containing protein kinase 1 (ROCK1) [77]. They suggested that miR-335 is down-regulated in various cancers as well as human HCC tissues and in four HCC cell lines [77]. These studies inform potential therapeutic strategies in HCC.

Lin et al. examined 19 known miRNAs that significantly increase in the sera of HCC patients and found that miR-210-3p is elevated in exosomes isolated from the sera of HCC patients [78]. Interestingly, exosomal miR-210 may derive from HCC cells, transfer into endothelial cells and promote tumor angiogenesis by inhibiting SMAD4 and signal transducer and activator of transcription 6 (STAT6).

Yu et al. identified five down-regulated miRNAs (miR-140-3p, miR-30d-5p, miR-29b-3p, miR-130b-3p and miR-330-5p), and one up-regulated miRNA (miR-296-3p) in the fast migrated HCC group compared to the slow migrated group by 372 HCC profiles from The Cancer Genome Atlas (TCGA) [79]. The targets of the six miRNAs were in the “focal adhesion” pathway and were consistent with the roles of tumor metastasis. Moreover, three miRNAs, miR-140, miR-30d and miR-29b, were significantly associated with patient survival. These exosomal miRNAs may be candidate biomarkers for predicting HCC cell migration and prognosis by the Kyoto Encyclopedia of Genes and Genomes (KEGG) and pathway analysis.

Shi et al. showed that exosome-delivered miR-638 in HCC is down-regulated and negatively associated with tumor size, vascular infiltration, TNM stage and overall survival [80]. They suggested that serum exosomal miR-638 may serve as a novel circulating biomarker and prognosis for HCC [80].

Liu et al. investigated that circulating HCC patient-derived exosomal miR-125b levels were down-regulated compared with those from patients with chronic hepatitis B and liver cirrhosis [81]. miR-125b has been reported to be a suppressor of HCC development through the inhibition of epithelial-mesenchymal transition, tumor growth, migration, and invasion of hepatoma cells. miR-125b directly regulates oncogenes, such as SMAD2/4, Sirtuin7 (SIRT7), suppressor of variegation 3-9 homolog 1 (SUV39H1), lin-28 homolog B (LIN28B), and phosphatidylinositol glycan anchor biosynthesis class F (PIGF) [81]. Moreover, miR-125b levels in exosomes were associated with tumor number, encapsulation, and TNM stage, and showed reduced time to recurrence (TTR) and overall survival (OS). These results indicated that exosomal miR-125b could serve as a promising prognostic marker for HCC.

Matsuura et al. found that HCC-derived exosomal miR-155 is up-regulated under hypoxic conditions [82]. Exosomal miR-155 induced tube formation in HUVECs, and high expression of exosomal miR-155 in preoperative plasma was significantly correlated with early recurrence.

Fu et al. showed that multidrug-resistant HCC cell-derived exosomal miR-32-5p is significantly elevated but PTEN is reduced. miR-32-5p activated the PI3K/Akt pathway by suppressing PTEN and by promoting angiogenesis and EMT causing multidrug resistance [8]. Clinically, elevated miR-32-5p and inhibited PTEN levels were positively associated with poor prognosis. 

Liu et al. showed that HCC-derived exosomal miR-25-5p was elevated and contributed to tumor self-seeding by enhancing cell migratory and invasive abilities in mouse xenograft models [83].

### 5.2. Long Noncoding RNA

Takahashi et al. found that HCC-derived exosomes enrich lncRNA-ROR [84]. A previous report showed that lncRNA-ROR expression was elevated in pluripotent stem cells and played a role in the derivation of pluripotent stem cells [85]. lncRNA-ROR was involved in the modulation of hepatoma cellular responses to sorafenib. Moreover, lncRNA-ROR increased during sorafenib treatment and played a functional role in chemoresistance by inhibiting sorafenib-induced cell death [84], supporting that therapeutic strategies targeting lnc-ROR could lead to enhance chemosensitivity in HCC. Similarly, lncRNA-VLDLR was enriched in HCC-derived exosomes and increased during anti-cancer treatment, such as sorafenib, camptothecin, and doxorubicin. lncRNA-VLDLR played a functional role in chemoresistance and its knockdown could lead to sorafenib-induced cell death [86]. A previous report [85] also showed that lncRNA-VLDLR expression is also elevated in embryonic stem cells.

Li et al. showed that lncRNA-FAL1 was up-regulated in HCC tissues and HCC-derived exosomes [87]. lncRNA-FAL1 accelerated cell proliferation and migration as a competing endogenous RNA (ceRNA) mechanism by competitively binding to miR-1236. miR-1236 is known to regulate hypoxia-induced EMT and cell metastasis through the inhibition of HDAC3 and SENP1 expression [88].

lnc-RNA may be useful as a novel diagnostic biomarker or a novel target for the treatment of HCC in the future. Hou et al. identified five prognostic lncRNAs as follows: CTD-2116N20.1, AC012074.2, RP11-538D16.2, LINC00501 and RP11-136I14.5, in HCC-derived exosomes [89]. They suggested that CTD-2116N20.1 and RP11-538D16.2 are correlated with a poor prognosis for HCC patients by regulating exosomal proteins. CTD-2116N20.1 is suggested to regulate proteins, such as cyclin B1 (CCNB1), cell division cycle-associated 3 (CDCA3), CDCA8, cyclin dependent kinase inhibitor 3 (CDKN3), E2F transcription factor 2 (E2F2), HAUS augmin like complex subunit 5 (HAUS5), lamin B2 (LMNB2), minichromosome maintenance complex component 4 (MCM4), MYB proto-oncogene-like 2 (MYBL2), polo-like kinase 1 (PLK1), RAD54-like (RAD54L), ribonucleotide reductase regulatory subunit M2 (RRM2), tubulin alpha 1b (TUBA1B) and nuclear receptor-binding SET domain protein 2 (NSD2/WHSC1), which are related to cell proliferation and tumor metastasis. RP11-538D16.2 modulates myosin XVI (MYO16) and glutamate-ammonia ligase (GLUL). MYO16 is involved in the cell cycle by regulating subcellular motor function and decreasing protein phosphatase catalytic activity [90]. GLUL affects oncogenesis in HCC [91].

Zhang et al. found that lncRNA-HEIH expression in both serum and exosomes increased in patients with HCV-related HCC [92]. However, the expression ratio in serum versus exosomes was decreased in patients with HCV-related HCC compared to patients with chronic hepatitis C.

Xu et al. investigated serum exosomal lncRNA ENSG00000258332.1 (LINC02394) and LINC00635 in experiments by comparing the sera between 55 HCC patients, 60 chronically HBV-infected patients and 60 healthy controls for the purpose of identifying potential diagnostic markers and prediction markers for the prognosis of HCC [93]. LINC00635 has been reported to predict the survival in patients with pancreatic ductal adenocarcinoma [94]. Both lncRNA ENSG00000258332.1 and LINC00635 levels in the HCC group were significantly higher than those in the other groups. Interestingly, a high ENSG00000258332.1 level in patients with HCC was associated with portal vein tumor thrombus, lymph node metastasis, TNM stage, and overall survival (OS), and a high LINC00635 level was related to lymph node metastasis, TNM stage, and OS. The combination of lncRNAs and serum AFP levels is useful for diagnosing HCC and predicting the prognosis of HCC.

Gramantieri et al. showed that lncRNA circulating cancer susceptibility 9 (CASC9) and lung cancer associated transcript 1 (LUCAT1) are up-regulated in HCC-derived exosomes [95]. LUCAT1 was demonstrated to directly target miR-181d-5p, which contributed to the stemness phenotype of hepatic progenitor cells and was elevated in an aggressive subgroup of HCC [96,97]. Higher CASC9 and LUCAT1 levels were associated with lower HCC recurrence after surgery, suggesting their potential usage as prognostic biomarkers for recurrence.

Sun et al. investigated eight candidates lncRNAs (LINC00462, CCAT1, CCAT2, HOTAIR, LINC00161, SPRY4-IT1, MALAT1 and UCA1) based on the available literature by quantitative reverse transcription-PCR (qRT-PCR) and determined that HCC-derived serum exosomal lncRNA-LINC00161 is up-regulated compared to that in healthy controls [98]. A previous report showed that LINC00161 is induced by cisplatin in osteosarcoma cells and sensitized osteosarcoma cells to cisplatin-induced apoptosis through the regulation of the miR-645-IFIT2 axis [99]. LINC00161 is also up-regulated in HCC urine samples and is sufficiently stable at different temperatures [98]. LINC00161 expression is significantly associated with serum AFP concentration and TNM stage, suggesting that exosome LINC00161 is a biomarker for the diagnosis of HCC.

Using qRT-PCR, Ma et al. found that lncRNA Jpx, which is an activator of X-inactive-specific transcript (Xist), was up-regulated in the exosomes of female HCC patients compared to healthy female volunteers and patients with chronic hepatitis B and cirrhosis [100]. lncRNA Xist was reported to affect cell proliferation and metastasis in HCC [101]. Exosomal lncRNA Jpx promoted Xist transcription by evicting CCCTC-binding factor (CTCF) on the Xist promoter. However, deleting the *Jpx* gene did not affect male cells, suggesting that *Jpx* was a sex-specific gene.

Li et al. found that the HCC-derived exosomal lncRNA TUC339 is up-regulated and was taken up by macrophages [102]. lncRNA TUC339 has been implicated in modulating HCC cell growth and adhesion [103]. Macrophages that take up lncRNA TUC339 increased pro-inflammatory cytokine production and enhanced M(IL-4) markers upon IFN-γ/LPS treatment, suggesting that lncRNA TUC339 is involved in the regulation of macrophage activation and M1/M2 polarization [102].

### 5.3. Messenger RNA (mRNA)

Abd El Gwad et al. found that lncRNA RP11-513I15.6 and miR-1262 are included in the RAB11A competing endogenous network [104]. These exosomal RNA-based biomarkers showed good sensitivity and specificity in distinguishing HCC from HCV and healthy controls. Additionally, RAB11A mRNA was the most independent prognostic factor. RAB11A has been reported to be up-regulated in non-small cell lung cancer and promotes proliferation, colony formation, invasion and migration with the up-regulation of cyclin D1 and cyclin E and with the down-regulation of p27 [105].

HCC patient-derived exosomal heterogeneous nuclear ribonucleoprotein H1 (hnRNPH1) was markedly higher than that in chronic hepatitis B patients and a healthy control [30]. A previous study demonstrated that hnRNPH1-mediated phosphorylation of phosphoribosyl pyrophosphate synthetase 1 (PRPS1) is required for HCC development [106]. The hnRNPH1 mRNA levels were associated with the Child-Pugh classification, portal vein tumor thrombus, lymph node metastasis, TNM stage and overall survival (OS). Moreover, the area under the ROC curve (AUC) for hnRNPH1 mRNA in combination with AFP was further improved compared to AFP only. These findings suggest that serum exosomal hnRNPH1 mRNA could be a useful marker for HCC in high endemic areas of HBV infection.

Wang et al. found that HCC patient-derived exosomal circular RNA PTGR1 (circPTGR1) was up-regulated compared to controls [107]. A previous study found that PTGR1 expression is regulated by nuclear factor (erythroid-derived-2)-like-2 (NRF2) and promotes HCC cell proliferation and antioxidant responses [108]. Elevated circPTGR1 promoted HCC metastasis by increasing migration and invasion via the miR449a-MET pathway. Moreover, the expression of exosomal circPTGR1 was associated with poor outcomes [108].

## 6. Exosome-Derived DNA

Exosomes are a distinct source of tumor DNA [109]. A higher percentage of patients with localized pancreatic ductal adenocarcinoma showed detectable KRAS mutations in exosome-derived DNA than previously reported for circulating cell-free DNA [109]. KRAS mutations were identified in 7.4%, 66.7%, 80%, and 85% of exosome-derived DNA and in 14.8%, 45.5%, 30.8%, and 57.9% of circulating cell-free DNA, in age-matched controls, localized, locally advanced, and metastatic PDAC patients, respectively [109]. As plasma circulating cell-free DNA levels were significantly higher in HCC patients than in non-HCC patients, it is possible that exosome-derived DNA may be useful as one of the diagnostic biomarkers for HCC [110].

## 7. Exosomes Are Potential Biomarkers and Potential Targets for the Treatment of HCC

Exosomes may be potential detection biomarkers for early-stage HCC (Table 1). Among exosomes, several good diagnostic markers have been reported [36,39,64,66,67,69,87,96]. Proteins, miRNAs and lncRNAs mediated by exosomes could provide new biomarker information for HCC (Table 1).

Exosomes may be potential biomarkers for predicting survival in HCC patients (Table 2) [8,30,35,45,46,47,73,79,80,81,82,85,89,93,94,95,107]. Exosomal circular RNA PTGR1 (circPTGR1) is up-regulated in HCC and the expression of exosomal circPTGR1 is associated with poor outcomes for patients with HCC [107]. 

Exosomes are also attractive targets for the treatment of HCC [10,11,41,42,43,102]. EIF3C is up-regulated during HCC tumor progression, and silencing EIF3C is known to induce cell apoptosis and suppress cell proliferation and tumor growth [9,37]. Thus, EIF3C is a potential target for treatment with exosome inhibitors or makes S100A11 reduction to suppress HCC angiogenesis and tumorigenesis. Stellate cell-derived exosomes can supply miR-335-5p cargo to recipient HCC cells and inhibit HCC cell proliferation and invasion [76,77]. miR-335 is down-regulated in various cancers as well as human HCC tissues, suggesting that miR-335 has therapeutic potential.

There is a substance that may also have the potential to treat HCC by changing the exosome contents. Xiong et al. found that exosomal miR-490 was up-regulated from mast cells stimulated by the HCV E2 envelope glycoprotein [111]. Moreover, miR-490 was up-regulated in recipient HepG2 cells by transferring the exosomal shuttle miR-490 into HCC cells, and it inhibits HCC cell metastasis by inhibiting the ERK1/2 pathway. This strategy may be a new biological therapy for HCV-associated HCC. Wei et al. found that vacuolar protein sorting 4 homolog A (Vps4A), which is a key regulator of exosome biogenesis, is frequently down-regulated in HCC [112]. They indicated that Vps4A increases the uptake of tumor suppressor miRNAs from exosomes in HCC cells, suggesting that Vps4A may exert a tumor-suppressive effect to regulate the secretion and uptake of exosomal miRNAs.

There are some reports that exhibit HCC therapeutic potential of HCC-derived exosomes [113,114,115]. Ko et al. showed that exosomes from adipose-derived mesenchymal stem cells suppress HCC growth by promoting antitumor responses of natural killer T (NKT) cells [113]. Li et al. found that recombinant adeno-associated viral vector (rAAV)-carrying AFP gene (*rAAV*/*AFP*)-transfected dendritic cells (DC)-derived exosomes (DEXs) stimulate naive T cell proliferation and induce T cell activation to become antigen-specific cytotoxic T lymphocytes (CTLs), exhibiting antitumor immune responses against HCC [114]. They suggested that DEX is promising for replacing mature DCs (mDCs) to function as cancer vaccines or natural antitumor adjuvants. Shi et al. showed that DCs with exosomes derived from tumor cells (DC-TEX) in combination with the PD-1 antibody increased the efficacy of sorafenib in HCC [115]. However, combining DC-TEX and sorafenib does not prolong patient survival, although DC-TEX decreases T-regulatory cells and increases CD8-positive T cells.

In contrast, there is a report that exhibits HCC resistance to sorafenib in HCC-derived exosomes. Qu et al. showed that HCC-derived exosomes induce sorafenib resistance by activating the HGF/c-Met/Akt signaling pathway, inhibiting sorafenib-induced apoptosis and elevating HGF, which may be an important mechanism underlying HCC resistance to sorafenib [116]. Serum exosome-delivered lncRNA-ROR, which is up-regulated in HCC, inhibited sorafenib-induced cell death and played a functional role in chemoresistance [84]. These results indicate that targeting lncRNA-ROR enhances chemosensitivity in HCC.

## 8. Potential Implications of Exosomes for Development of Therapeutic Strategies to Treat HCC

Trivedi et al. reported that miRs in the exosomesare related to the activation genes associated with anti-tumor signaling [117]. These affect not only the cancer cells but also the tumor microenvironment by using the “bystander effect” through genetic transfer with secreted exosomes. Transfer of exosomes reduces the cardiotoxicity of anti-tumor therapies with doxorubicin [118]. Exosomes are useful as drug delivery tools because of exosomes resembling liposomes [119,120]. Thus, transfer of exosomes is one of the therapeutic strategies for clinic use to treat HCC. Exosome-based drug delivery approaches in preclinical and clinical trials have been demonstrated to dramatically inhibit cancer development [119].

## 9. Conclusions

Exosomes mediate the communication and the transfer of several molecules implicated in exosome biogenesis between both HCC and non-HCC cells involved in tumor-associated cells, and thereby are both tumorigenic and tumor suppressors. It is difficult to diagnose early-stage HCC and to treat HCC radically. HCC-derived exosomes may be useful for the diagnosis of HCC as novel HCC biomarkers and for the treatment of HCC as therapeutic targets and as treatment tools via representing the major delivery system for proteins and several types of RNAs. The limitation of this review is that we did not completely incorporate the current findings and concept of exosomes in the various liver diseases that are associated with the formation of HCC because the development of HCC originates from the long-term disease progression and leading force of liver-associated diseases. Most studies emphasized the implications of exosomes as biomarkers in diagnosis of human diseases. However, exosomes have diverse functions in the maintenance of homeostasis. Whether and how exosomes sever as reliable and practicable biomarkers remain largely controversial [121]. These fields are interesting and need to be further studied.

## Figures and Tables

**Figure 1 ijms-20-01406-f001:**
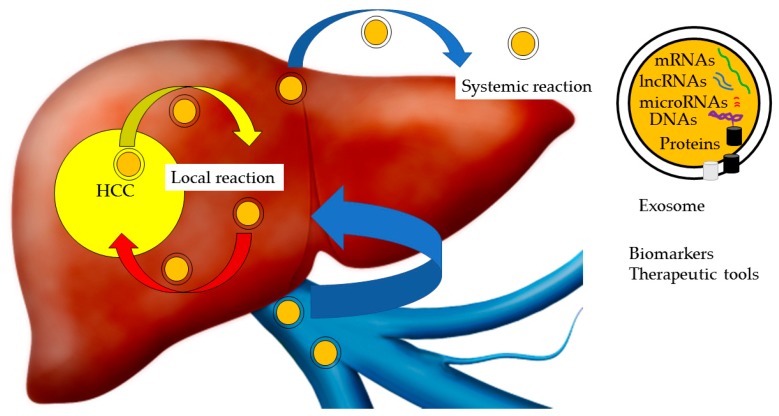
Exosomes and hepatocellular carcinoma (HCC). Exosomes transfer proteins, DNAs and various forms of RNA, such as microRNA (miRNA), long noncoding RNA (lncRNA) and messenger RNA (mRNA), between HCC and non-HCC cells. Exosomes induce local and systemic reactions, contributing to the development of HCC. Exosomes are useful biomarkers and therapeutic tools for HCC.

**Table 1 ijms-20-01406-t001:** Exosomal biomarkers for the detection of hepatocellular carcinoma.

Exosomal Markers	HCC	Actions	References
CAP1	Up-regulation	Potential metastasis	[39]
S100A4	Up-regulation	Modulating the cytoskeleton proteins	[36]
miR-122	Down-regulation	Carcinogenesis	[64]
miR-21	Up-regulation	Cell migration and invasion	[66,69]
miR-519d	Up-regulation	Cell proliferation, invasion and anti-apoptosis	[67]
lncRNA-FAL1	Up-regulation	Cell proliferation and migration	[87]
LINC00161	Up-regulation	Chemoresistance	[98]

CAP1, cyclase associated actin cytoskeleton regulatory protein 1; S100A4, S100 calcium binding protein A4; Gal-3BP, galectin-3-binding protein; PIGR, polymeric immunoglobulin receptor.

**Table 2 ijms-20-01406-t002:** Exosomal biomarkers for the prediction of survival of patients with hepatocellular carcinoma.

Contents of Exosomes	Molecules	Actions	References
Proteins	SMAD3	Potential metastasis	[35,45,46,47]
	14-3-3ζ	EMT	
	S100A11	Angiogenesis	
MicroRNAs	miR-29b-3p	Cell migration	[8,73,79,80,81,82]
	miR-30d-5p	Cell migration	
	miR-32-5p	Multidrug resistance	
	miR-125b	Tumor suppressor for HCC	
	miR-140-3p	Cell migration	
	miR-155	Angiogenesis	
	miR-638	EMT, invasion	
	miR-718	Suppression proliferation and HOXB8	
Long noncoding RNA	CASC9	Cell proliferation	[89,93,94,95]
	LUCAT1	Proliferation and metastasis	
	ENSG00000258332.1	Portal vein tumor emboli and lymph node metastasis	
	LINC00635	Lymph node metastasis	
	CTD-2116N20.1	Poor prognosis	
	RP11-538D16.2	Poor prognosis	
Circular RNA	circPTGR1	Migration and invasion	[107]
Messenger RNA	hnRNPH1	Portal vein tumor emboli and lymph node metastasis	[30,46,47]
	14-3-3ζ	EMT	

SMAD3, SMAD family member 3; EMT, Epithelial-mesenchymal transition; S100A11, S100 calcium binding protein A11; HOXB8, homeobox B8; hnRNPH1, heterogeneous nuclear ribonucleoprotein H1.

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
