# Peer review of "Exosomes and Hepatocellular Carcinoma: From Bench to Bedside"

_ijms, 2019, doi:10.3390/ijms20061406_

Reviewer 1 Report

Sasaki et al. reviewed the current knowledge of exosomes and the roles of exosomes-associated components in the development of hepatocellular carcinoma. This manuscript provided a new frame of the physiological significance of exosomes in the disease progression of liver cancer as well as the potential implications of exosomes as biomarkers for detection and therapeutic targeting to cure hepatocellular carcinoma. Although this manuscript review the most studies of roles of exosomes in hepatocellular carcinoma, the current version of this manuscript remains several points that must be improved before publication in IJMS. 

Point #1:

Several sentences in the context of manuscript seem to be nearly identical to those in abstract. For example, sentences in lines 35 and 36 in the section of introduction are nearly identical to those in lines 17 and 18 in "abstract". The authors should be aware of duplication in writing manuscript.

Point #2:

The authors stated the "hepatocellular carcinoma largely occurs in the background of chronic liver disease and cirrhosis" in lines 14 and 15 of abstract. In general, liver cirrhosis is a late-staged form of liver diseases. Please explains the concern for writing this statement. 

Point #3: 

Several references in the section of introduction were not appropriately listed in this version of manuscript. For example, ref. 10-30 for the current treatment and development of hepatocellular carcinoma were not correctly and comprehensively reviewed. Particularly, there were several mistakes in referring the literature for controlling HBV and HCV infection by vaccination and DAAs. 

Point #4:

Since the development of hepatocellular carcinoma originates from the long-term disease progression and leading force of liver-associated diseases, the authors must incorporate the current findings and concept of exosomes in the various liver diseases the are associated with the formation of hepatocellular carcinoma.

Point #5:

Most content in this manuscript for roles of exosomes-associated molecules in the development of hepatocellular carcinoma focused on the "description" on the findings in the previous studies. The detailed roles of exosomes in hepatocellular carcinoma were not precisely consolidated and organized in the current version of this manuscript. The authors must largely improve the organization and conclusion of current findings on the relationship between exosomes and hepatocellular carcinoma. 

Point #6:

The roles of exosomes in the hepatocellular carcinoma in each sections must be consolidated in tables. For example, there should be a table that summarizes the role of exosomal proteins in the occurence of hepatocellular carcinoma.

Point #7:

Most studies emphasized the implications of exosomes as biomarkers in diagnosis of human diseases. However, exosomes have diverse functions in the maintenance of homeostasis. Whether and how exosomes sever as reliable and practicable biomarkers still remain largely controversial. The authors must discuss the possible hurdles for the implications of exosomes in the detection of hepatocellular carcinoma. 

Point #8:

The potential implications of exosomes for development of therapeutic strategies for clinic use to treat hepatocellular carcinoma are not exactly discussed in this version of manuscript. The authors must improve this section. 

Point #9:

The author must be aware of citations in references section. Several inappropriate and incorrect references were listed in this version of manuscript. The authors must improve this points. 

Point #10:

There were several grammatical errors and incorrect wording for scientific presentation. The authors must improve this points.

Author Response

Response to reviewer #1: Thank you for your valuable comments and criticisms.

Response to your comment #1: “Several sentences in the context of manuscript seem to be nearly identical to those in abstract. For example, sentences in lines 35 and 36 in the section of introduction are nearly identical to those in lines 17 and 18 in "abstract". The authors should be aware of duplication in writing manuscript.”

Thank you for your valuable comment. We agree with you. Accordingly, we revised our manuscript as follows.

In page 1, lines 34-38,

…numerous biological processes in both normal and pathologic contexts [2]. Evidence indicates that proteins, DNAs and various forms of RNA, such as microRNA (miRNA), long noncoding RNA (lncRNA) and messenger RNA (mRNA) transferred by exosomes, contribute to the development of cancers, neurodegenerative disorders, collagen diseases and other pathological conditions (Figure 1) [3-6]. Nonetheless…

In page 9, lines 24-25,

several good diagnostic markers have been reported [36, 39, 64, 66, 67, 69, 87, 96]. Proteins, miRNAs and lncRNAs mediated by exosomes could provide new biomarker information for HCC (Table 1).

In page 10, line 3,

Exosomes are also attractive targets for the treatment of HCC [10,11,41-43,102]. EIF3C is…

Response to your comment #2: “The authors stated the "hepatocellular carcinoma largely occurs in the background of chronic liver disease and cirrhosis" in lines 14 and 15 of abstract. In general, liver cirrhosis is a late-staged form of liver diseases. Please explains the concern for writing this statement.”

Thank you for your valuable comment. We agree with you. Accordingly, we revised our manuscript as follows.

In page 1, lines 14-16,

Abstract: As hepatocellular carcinoma (HCC) usually occurs in the background of cirrhosis, which is an end-stage form of liver diseases, treatment options for advanced HCC are limited, due to poor liver function. The exosome is…

Response to your comment #3: “Several references in the section of introduction were not appropriately listed in this version of manuscript. For example, ref. 10-30 for the current treatment and development of hepatocellular carcinoma were not correctly and comprehensively reviewed. Particularly, there were several mistakes in referring the literature for controlling HBV and HCV infection by vaccination and DAAs.”

Thank you for your valuable comment. We agree with you. Accordingly, we revised our manuscript as follows.

In page 2, line 1-page 3, line 2,

Hepatocellular carcinoma (HCC) is the fourth most common cancer type and the second most common cause of cancer-related deaths [12]. HCC is mainly associated with chronic liver disease and cirrhosis [13]. Recently, therapeutic options for hepatitis C virus (HCV) and hepatitis B virus (HBV), which are responsible for chronic liver diseases, have been developed such as direct-acting antiviral agents (DAAs) and nucleos(t)ide analogues, respectively [14, 15]. Universal HBV vaccination has also been introduced in many countries [15]. However, obesity leading to non-alcoholic liver disease (NAFLD) and non-alcoholic steatohepatitis (NASH) is increasing as one of the major etiologies for liver diseases and HCC [16]. The treatment options for advanced-stage HCC are very limited. HCC can be treated with liver transplantation, surgical resection, radiofrequency ablation (RFA), transcatheter arterial chemoembolization (TACE) and systemic chemotherapy [13]. Among these options, liver transplantation and surgical resection seem to be potential curative approaches. However, some patients drop out from liver transplantation because of the increasing waiting time and the shortage of donor organs [13].

This review focuses on the contents of exosomes and how exosomes contribute to the development of HCC. We further address how useful exosomes are to diagnose HCC, to treat patients with HCC and to predict their prognosis.

2. HCC

HCC largely occurs in the background of chronic liver disease and cirrhosis in over 90% of cases [13]. In the US and Japan, HCV is the major cause of chronic hepatitis, cirrhosis, and HCC, while in Asian and African countries, HBV is a common cause for these diseases [15]. Other etiologies, such as alcoholic liver disease, NAFLD and NASH, also contribute to chronic liver diseases and HCC. Recently, DAAs have been shown to eradicate HCV from over 90% of chronically HCV-infected patients [14]. As of 2012, 183 countries initiated an HBV vaccination program for newborn infants, which decreased HBV-associated liver diseases [15]. However, obesity leading to NAFLD and NASH is increasing as one of the leading etiologies for liver diseases and HCC [17].

The high mortality rates of HCC are almost equal to the incidence rates of HCC in most countries, indicating the lack of effective therapies [13]. The treatment options for advanced HCC are very limited, although there are several treatment options for HCC as follows: liver transplantation, surgical resection, RFA, TACE and systemic chemotherapy [13,18]. Moreover, the recurrence rate of HCC after surgical resection is 30-70% within 5 years [19-21].

Response to your comment #4: “Since the development of hepatocellular carcinoma originates from the long-term disease progression and leading force of liver-associated diseases, the authors must incorporate the current findings and concept of exosomes in the various liver diseases the are associated with the formation of hepatocellular carcinoma.”

Thank you for your valuable comment. We agree with you. Accordingly, we revised our manuscript as follows.

In page 11, lines 21 – 28,

…and several types of RNAs. The limitation of this review is that we did not completely incorporate the current findings and concept of exosomes in the various liver diseases that are associated with the formation of HCC because the development of HCC originates from the long-term disease progression and leading force of liver-associated diseases. Most studies emphasized the implications of exosomes as biomarkers in diagnosis of human diseases. However, exosomes have diverse functions in the maintenance of homeostasis. Whether and how exosomes sever as reliable and practicable biomarkers remain largely controversial [121]. These fields are interesting and need to be further studied.

Response to your comment #5: “Most content in this manuscript for roles of exosomes-associated molecules in the development of hepatocellular carcinoma focused on the "description" on the findings in the previous studies. The detailed roles of exosomes in hepatocellular carcinoma were not precisely consolidated and organized in the current version of this manuscript. The authors must largely improve the organization and conclusion of current findings on the relationship between exosomes and hepatocellular carcinoma.”

Thank you for your valuable comment. We agree with you. Accordingly, we revised Table 2 of our manuscript.

Response to your comment #6: “The roles of exosomes in the hepatocellular carcinoma in each sections must be consolidated in tables. For example, there should be a table that summarizes the role of exosomal proteins in the occurence of hepatocellular carcinoma.”

Thank you for your valuable comment. We agree with you. Accordingly, we revised Table 2 of our manuscript.

Response to your comment #7: “Most studies emphasized the implications of exosomes as biomarkers in diagnosis of human diseases. However, exosomes have diverse functions in the maintenance of homeostasis. Whether and how exosomes sever as reliable and practicable biomarkers still remain largely controversial. The authors must discuss the possible hurdles for the implications of exosomes in the detection of hepatocellular carcinoma.”

Thank you for your valuable comment. We agree with you. Accordingly, we revised our manuscript as follows.

In page 11, lines 21 – 28,

…and several types of RNAs. The limitation of this review is that we did not completely incorporate the current findings and concept of exosomes in the various liver diseases that are associated with the formation of HCC because the development of HCC originates from the long-term disease progression and leading force of liver-associated diseases. Most studies emphasized the implications of exosomes as biomarkers in diagnosis of human diseases. However, exosomes have diverse functions in the maintenance of homeostasis. Whether and how exosomes sever as reliable and practicable biomarkers remain largely controversial [121]. These fields are interesting and need to be further studied.

Response to your comment #8: “The potential implications of exosomes for development of therapeutic strategies for clinic use to treat hepatocellular carcinoma are not exactly discussed in this version of manuscript. The authors must improve this section.”

Thank you for your valuable comment. We agree with you. Accordingly, we revised our manuscript as follows.

In page 11, lines 6-14,

8. Potential implications of exosomes for development of therapeutic strategies to treat HCC

Trivedi et al. reported that miRs in the exosomesare related to the activation genes associated with anti-tumor signaling [117]. These affect not only the cancer cells but also the tumor microenvironment by utilizing the 'bystander effect' through genetic transfer with secreted exosomes. Transfer of exosomes reduces the cardiotoxicity of anti-tumor therapies with doxorubicin [118]. Exosomes are useful as drug delivery tools because of exosomes resembling liposomes [119,120]. Thus, transfer of exosomes is one of the therapeutic strategies for clinic use to treat HCC. Exosome-based drug delivery approaches in preclinical and clinical trials have been demonstrated to dramatically inhibit cancer development [119].

Response to your comment #9: “The author must be aware of citations in references section. Several inappropriate and incorrect references were listed in this version of manuscript. The authors must improve this points.”

Thank you for your valuable comment. We agree with you. We revised our manuscript accordingly.

Response to your comment #10: “There were several grammatical errors and incorrect wording for scientific presentation. The authors must improve this points.”

Thank you for your valuable comment. We agree with you. We revised our manuscript accordingly.

Reviewer 2 Report

Concerning the manuscript IJMS-455316 by Saski et al which is a review on the role of the exosomes in hepatocarcinoma biology, progression and possible uses in diagnosis, prognosis and therapy.

It is extremely well written and presented and is an important contribution for who wishes to have a consise review on this important topic. Importantly, the English is correct and consise which greatly helps the reading and understanding.

Furthermore, the organization is perfect and makes for the easy progression from point to point to the final conclusions.

I thank the journal for giving me the opportunity to review this nice contribution to the field.

I believe that this solid review warrents being immediately published.

Author Response

Response to reviewer #2: Thank you for your valuable comments and criticisms. Thank you very much for your encouraging comments.

Round  2

Reviewer 1 Report

The content of this revised manuscript has been strengthened and well-prepared for publication in IJMS.